# From Double-Strand Break Recognition to Cell-Cycle Checkpoint Activation: High Content and Resolution Image Cytometry Unmasks 53BP1 Multiple Roles in DNA Damage Response and p53 Action

**DOI:** 10.3390/ijms231710193

**Published:** 2022-09-05

**Authors:** Laura Furia, Simone Pelicci, Mirco Scanarini, Pier Giuseppe Pelicci, Mario Faretta

**Affiliations:** 1Department of Experimental Oncology, European Institute of Oncology IRCCS, 20139 Milan, Italy; 2Department of Oncology and Hemato-Oncology, University of Milan, 20122 Milan, Italy

**Keywords:** DNA damage response, 53BP1, p53, cell-cycle, fluorescence microscopy, image analysis, image cytometry

## Abstract

53BP1 protein has been isolated in-vitro as a putative p53 interactor. From the discovery of its engagement in the DNA-Damage Response (DDR), its role in sustaining the activity of the p53-regulated transcriptional program has been frequently under-evaluated, even in the case of a specific response to numerous DNA Double-Strand Breaks (DSBs), i.e., exposure to ionizing radiation. The localization of 53BP1 protein constitutes a key to decipher the network of activities exerted in response to stress. We present here an automated-microscopy for image cytometry protocol to analyze the evolution of the DDR, and to demonstrate how 53BP1 moved from damaged sites, where the well-known interaction with the DSB marker γH2A.X takes place, to nucleoplasm, interacting with p53, and enhancing the transcriptional regulation of the guardian of the genome protein. Molecular interactions have been quantitatively described and spatiotemporally localized at the highest spatial resolution by a simultaneous analysis of the impairment of the cell-cycle progression. Thanks to the high statistical sampling of the presented protocol, we provide a detailed quantitative description of the molecular events following the DSBs formation. Single-Molecule Localization Microscopy (SMLM) Analysis finally confirmed the p53–53BP1 interaction on the tens of nanometers scale during the distinct phases of the response.

## 1. Introduction

Among the wide variety of threats to the integrity of the cell genome, DNA Double-Strand Breaks (DSBs) are the most toxic and dangerous. The cells developed a complex molecular network, the DNA Damage Response (DDR) machinery, able to orchestrate the response to recognize and fix them [1,2,3,4,5,6].

Upon DSB recognition by the MRE11–Rad51–NBS1 (MRN) complex, a signaling cascade starts to simultaneously recruit molecules to repair the damage and to trigger a cell-cycle arrest and, eventually, a commitment to senescence or apotosis, in case DNA breaks could not be properly fixed. DSBs are repaired by two distinct pathways: classical Non-Homologous End-Joining (c-NHEJ) and Homology-Directed Repair (HDR) [1,7,8].

The p53 Binding Protein 1 (53BP1) is a key regulator in the choice of the DSB repair pathway, and plays a pivotal role in promoting c-NHEJ [9,10,11]. Originally, 53BP1 was identified alongside ASPP2 (53BP2) as a yeast two-hybrid interactor of the p53 DNA binding domain [12]. Even if p53 acts as a key regulator in determining the duration of the cell-cycle arrest and the definitive cell fate, the interaction between the two proteins was not frequently investigated. Even if 53BP1 stimulates p53-mediated transcriptional activation in-vitro [13], 53BP1’s role in DD recognition and in the choice of the DNA repair pathway obscured, for a long time, the interest on the putative support that 53BP1 can exert in-vivo on the p53-induced selective activation of targeted gene expression.

Recent works [14,15,16,17] highlighted a fundamental role of 53BP1 in the p53 control of cell-fate programs.

Molecular spatiotemporal relocalization is one of the main mechanisms the cell employs to control and coordinate processes: 53BP1 could, thus, exert different spatially-modulated activities in DSBs foci and p53-targeted genomic loci.

Investigating the potential link among the intracellular position and function requires a sufficiently high spatial resolution. Widefield and Confocal Fluorescence Microscopy and their super-resolving derivatives (i.e., Stimulated Emission Depletion Microscopy (STED); and Single Molecule Localization Microscopy (SMLM) with its variants (e.g., Stochastic Optical Reconstruction Microscopy (STORM), PhotoActivated Localization Microscopy (PALM)) offer the possibility to locate molecules in a range from 200 to 20 nm in single cells [18,19,20,21]. However, the increase in spatial resolution is always associated with a low statistical sampling due to the progressively restricted field of view. High resolution widefield fluorescence microscopy requires the employment of numerical aperture objectives that maximize the amount of collected photons with an acceptable linear extension of the imaged region. Unfortunately, this high sensitivity and signal-to-noise ratio were seldom employed to quantify the abundance of the targeted molecules.

To overcome these limitations, we developed an image–cytometry pipeline in the framework of automated fluorescence microscopy [22,23,24,25]. The developed protocols and computational tools allow the coupling of a diffraction limited imaging to a statistical sampling in the order of several thousands of cells. Moreover, the optimization of the staining protocols allows reaching the simultaneous evaluation of up to seven fluorescence channels.

Here, we employed the developed tools to quantitatively describe the DNA Damage Response (DDR) in cells exposed to X-ray Ionizing Radiation (IR). We coupled the statistical sampling and resolution of a diffraction limited analysis to a “pulse and chase” protocol inherited from flow cytometry. We, thus, obtained a detailed quantitative description of the differential effects of DNA Damage (DD) induction on the different cell populations according to the cell-cycle phase at the moment of the irradiation. The approach has the value of avoiding potential artefacts induced by additional synchronizing agents.

The possibility to retrieve cells of interest after the first acquisition was then employed to perform a 3D high-resolution image collection of specific phenotypes of interest identified by the image cytometry analysis. Finally, the interactions among the component of the DDR and checkpoint-activation machineries were targeted at a spatial scale less than about 50 nm by Proximity Ligation Analysis (PLA) [26,27,28] and SMLM.

Here, we validate the developed pipeline, showing that, in response to irradiation, the interplay among DNA damage recognition, repair, and cell-cycle control led to a spatiotemporally modulated interaction between the 53BP1 DDR protein and p53 checkpoint activator.

## 2. Results and Discussion

### 2.1. Kinetics of X-rays Irradiation Induced DNA Damage, Processing, and Cell-Cycle Arrest

To dissect the interplay among DNA-Damage Recognition (DDR) and cell-cycle progression control, we employed an image cytometry pipeline based on an automated fluorescence microscope [22,23,24,25]. The pipeline associates a high-resolution seven-color fluorescence microscopy analysis (i.e., DNA, EdU, p53, p21, KI67, γH2A.X, 53BP1) (representative pictures from the analysis in Appendix A) to a statistical sampling of several thousands of cells. The cell response after DD induction has been characterized by measuring the content of markers of the activated checkpoint (p53 and p21) and of the proliferation status (KI67). A “pulse and chase” protocol marking initial S-phase cells allowed monitoring of the cell-cycle perturbation [29]: exponentially growing cells have been incubated with the Uridine analogue, EdU, for a brief time to mark active DNA synthesis. Then, fresh medium has been added to replace the one containing the synthetic nucleotide to let the cells grow. Only cells actively synthetizing DNA during the incubation were, thus, marked by EdU positivity, and the simultaneous evaluation of EdU signal and DNA stain allowed to follow the cell-cycle progression of the cells irradiated during the G1, S, and G2M phases. The approach indicated, in 13–15 h, a precise estimation of the cell-cycle duration of MCF10A non-transformed breast epithelial cells, whereas DNA synthesis requires 4–5 h to be completed (Figure 1). We, thus, defined a series of time-points to monitor the effects of a 5-Gy X-ray irradiation on the first cell-cycle: its fast dynamics were dramatically altered with differential effects on the cells according to the phase at the time of exposure to the radiation. G1 to S phase progression occurred only for a fraction of cells, revealing the activation of the G1 checkpoint that blocked the transition that took place between 3 and 6 h under unperturbed exponential growth. At the same time-point, the absence of newly divided cells among the EdU positive population showed the simultaneous induction of a G2 arrest. DNA replication was not arrested, but dramatically slowed down: usually, from 3 to 6 h, all the exponentially growing cells reached the G2 phase, whereas cells irradiated during DNA synthesis required 9 to 12 h to complete the replication process before being arrested in G2. At the end of the first cell-cycle, the entire population was blocked in the G1 and G2 phases, and the block was still maintained 24 h after the irradiation.

Being the cells immediately fixed after exposure to X-rays, no changes in the protein content distribution in the cell-cycle have been detected (Appendix A).

Measurement of the p53 amount per cell confirmed that its accumulation was only required for activation of the checkpoint without a concurrent stabilization: p53 content peaked at 3 h, but returned close to basal level within the next time-point, 3 h later (Figure 1, panel B). Later, between 12 and 24h, when, in exponential growth, a second replication cycle occurred, some of the blocked cells exhibited p53 stabilization, increasing their content over the unperturbed level.

Maintenance of the G1 and G2 phase arrest was instead granted by a strong increase in the amount of p21 protein present in the blocked cells (Figure 1, panel B). Strikingly, the cytostatic effect of the irradiation did not cause an exit from the cell-cycle. Analysis of the KI67 proliferation marker revealed that, at all the examined time-points, the cells maintained a KI67 content comparable to the initial one, independently from their arrested cell-cycle progression (Figure 1, panel B). We only observed a temporary reduction at 6 h that was recovered in the next hours, mainly restricted to the G1 phase.

Analysis of the temporal evolution of the γH2A.X and 53BP1 Ionizing-Radiation (IR)-induced foci provided the picture of the DDR response during the first cell-cycle, and up to 24 h after irradiation (Figure 2, panel A). The integrated intensity of the total number of foci per nucleus has been selected as best measurement to quantify the induced damage [25]. Thanks to the diffraction-limited spatial resolution of the collected images, it is also possible to quantify the number and size of the detected foci to obtain a detailed description of the DDR dynamics over the entire population.

As expected, 5-Gy irradiation immediately activated DDR by phosphorylating H2A.X: the increase in the signal, proportional to the amount of induced DD, is dramatically higher than the endogenous damage, associated to the replication stress during the DNA synthesis, and proportional to the content of DNA per cell, as shown by the progressive increase across the cell-cycle phases. As time went on, with DNA repair taking place, the average level was constantly reduced until reaching a minimal, but still consistent, value 24 h after exposure to IR. This reduction was not limited to the G1 and G2 arrested cells, but it was also exhibited by actively replicating cells, suggesting that the marked slowed down in the S phase completion could be attributed to the ongoing DNA replication and parallel processing of the DNA damage. The number of foci followed the same kinetics exhibited by the integrated intensity of γH2A.X per cell, with a progressive decline with time. However, the evaluation of average size per cell instead showed a growing trend, indicating that the reduction in foci number was not attributed to their disappearance, but to their fusion in larger entities in agreement with a model of chromatin movement for DSB clustering [30,31,32].

The kinetics of 53BP1 foci evidenced an opposite behavior in agreement with the double role of the protein in DD recognition and repair (Figure 2, panel B). Differently from the progressive reduction in intensity of the γH2A.X foci, the spatial distribution of 53BP1 into foci was maintained and surprisingly increased from the first instants after irradiation up to 12 h. The temporal progression followed an increase throughout all of the first cell-cycle, concentrating the augmented signal into the G1 and G2 phases; after initial recruitment, S phase cells did not further accumulate 53BP1. The evolution of the number and size of foci paralleled the behavior observed for γH2A.X foci, further reinforcing the hypothesis of clustering to repair DNA.

The observed distributions provided a perfect picture of the ongoing molecular events: 53BP1 re-localized to foci to sustain DD recognition and to activate DD response. However, its role in damage processing and repair was not exerted during replication, where homologous recombination took place, requiring, instead, its increasing presence to support non-homologous end joining in the arrested cells. As further evidence, we analyzed the DDR foci population independently from the cell of origin: the statistical distribution of the signal generated by H2A.X phosphorylation and 53BP1 recruitment perfectly overlapped the above described steps. The initial damage generated foci where the modified histone did not yet accumulate 53BP1 (Figure 2 panel C, 0 h), as evidenced by the cloud spread parallel to the *X* axis. The ongoing recruitment due to DD recognition caused the distribution of events along the bisector, whereas the vertical cloud of intensity values corresponding to 53BP1-enriched foci was reminiscent of the repair of pre-irradiation endogenous damage, where the disappearance of 53BP1 was delayed with respect to the phosphorylated histone marker. The mean intensity of the γH2A.X in foci was progressively reduced in favor of a massive increase in the 53BP1 fluorescence that maintained its maximum levels from 12 to 24 h.

The measurement of the total amount of 53BP1 protein in the cell nuclei showed, in parallel, a similar increase (Figure 3). However, the fraction of the intensity generated by the protein localized in foci over the one linked to the fluorescence of the entire nucleus shed light on a completely new distribution. Only the first time-points, from immediately after irradiation up to 6 h later, exhibited a constant and maximized fraction of signal in the foci. From 9 h, a growing population of cells appeared where the amount of protein present in the nucleoplasm acquired relevance. Even if the 53BP1 intensity in foci was still high and growing in this time interval, the high amount of protein detected in cell nuclei and its massive localization in the nucleoplasm may be attributed to its requirement for additional functions, and not solely to an involvement in DDR and repair.

### 2.2. Kinetic High-Resolution Analysis of the Interactions among DDR and Checkpoint Molecular Networks

53BP1 was first isolated as a p53 interactor in an in-vitro two-hybrid screening [12]. After the discovery of its role in DDR, its in-vivo putative function in the regulation of p53 protein was somehow dimmed until the discovery of 53BP1’s ability to bind methylated proteins involved in cell-cycle control [33], and to support p53 activity in determining cell fate with a mechanism distinct from DNA recognition [14,16]. The simultaneous accumulation of p53 and 53BP1 at late time-points and the increased fraction of nucleoplasm-diffused 53BP1 molecules led us to explore their putative interaction, simultaneously monitoring the other molecular events that could potentially take place in response to IR exposure. Considering that 24 h after the irradiation, cells were still arrested almost at the end of two cell-cycles, we extended the observation period to 48 h to highlight the cell fate of the blocked populations, a time where p53 exerted a fundamental role in the regulation of the occurring events.

According to the bivariate distribution of DNA versus EdU content, six different subpopulations have been identified to monitor the progression through the phases of the cell-cycle. The average contents per cell of the seven above-mentioned parameters were, thus, calculated over repeated experiments to quantify their variations (Figure 4).

The measured values perfectly reproduced the behavior observed in the qualitative analysis of the first cell-cycle. p53 showed a transitory peak in its content before being stabilized, starting from 24 h. This stabilization perfectly correlated with the appearance of cell mortality, with several micronuclei detected 48 h after irradiation (Appendix A).

The other checkpoint protein, p21, after the initial strong accumulation required to immediately arrest the cell-cycle progression, showed a partial, but progressive, reduction of content in the arrested cells. In parallel, the KI67 proliferation marker instead exhibited an increase mainly concentrated in the G2-stopped fraction. Strikingly, the two events correlated not only with the setting of a potential recovery of the growth, but also with the simultaneous onset of cell death. The impossibility to maintain a permanent cell arrest could, thus, lead to a release of the block by degrading p21 with the simultaneous synthesis of the KI67 protein to favor cell mitosis, inevitably leading to an aberrant and lethal conclusion due to the massive presence of DD, as marked by the appearance of micronuclei.

The total amount per cell of the 53BP1 protein, though stable during the first cell-cycle after irradiation, progressively increased. However, a simultaneous switch in its intracellular localization occurred, leading to the protein accumulation in the nucleoplasm, with a consequent decrease in the fraction associated to DSBs. At the same time, p53 was stabilized, sustaining a model where 53BP1 was requested to support p53 activity [14] with a function distinct from DDR, which was instead dramatically reduced in the late time-points. The quantification of the DD-describing parameters, namely intensity, number, and area of the IR-induced foci, points towards a resolution and clusterization of the damage within 24 h from the irradiation.

Considering that some of the targeted molecules showed a complete (p53, p21) or a partial (53BP1), but consistent, diffused localization in the nucleoplasm, the detection and localization of putative interactions is strongly limited even at the diffraction limit of 200 nm.

To gain resolution at the molecular scale without compromising the high statistical sampling (at least 5000 cells analyzed per each time-point, from 10^4^ to 10^6^ detected foci) granted by the developed image cytometry approach, we performed a Proximity Ligation Analysis (PLA) [25,26,27,28]. We focused on the time-points corresponding to the peaks in p53 content to evaluate the putative interactions between the components of the DDR and checkpoint-activating machinery. The first analysis was performed 6 h after the irradiation to catch the initial increment of the guardian of the genome, in coincidence with the highest activity of the DDR. Then, we moved to 24–48 h after irradiation to examine the behavior of 53BP1 when p53 accumulated to regulate cell fate while IR-induced DSBs were almost completely processed.

Though 53BP1 recruitment to IR foci provided a positive control of the PLA (Appendix A), the analysis of 53BP1 interactions with p53 showed a more variable scenario. The first peak of expression of p53 coincided with an increase in the number of detected PLA foci independently of the cell-cycle position (Figure 5 Panel A). At 6 h, when the DDR activity caused a massive recruitment of 53BP1 into the IR foci, a partial correlation between the interaction spots number and the amount of damage was detected: when only PLA spots adjacent to DD foci were considered, the intensity of 53BP1 foci and the average number of PLA spots in nuclei exhibited a linear trend in their distribution (Pearson coefficient = 0.72; average number of PLA spots = 5.7 over 16.2) (Figure 6 Panel B). Twenty-four hours after irradiation, the correlation disappeared, as witnessed by the spreading of the events along the horizontal axis. Despite an increase in the average number of interaction spots, the drop in their fraction adjacent to DDR foci (average number = 2.4 over 17.7) witnessed a shift of the putative 53BP1–p53 complex towards the nucleoplasm.

The classification of the cells according to the magnitude of the analyzed interaction confirmed that the correlation detected 6 h after the irradiation was limited only to a fraction of the PLA spots. Cells distributed above or below the median value of the number of PLA spots per cell, calculated over the entire population, exhibited similar levels of damage, measured as the amount of 53BP1 recruited into IR foci (Figure 5, Panel C). The total amount of 53BP1 protein in the nucleus did not greatly influence the p53–53BP1 interaction, which appeared to be mainly sensible to the p53 content. The analysis of the late time-points confirmed no differences detected in the DDR response when cells were ordered on the base of the enrichment in the interaction spots, suggesting that the formation of a 53BP1–p53 complex was an event only partially dependent on the DNA damage.

It could be argued that the detected nucleoplasmic spots may result from the increased molecular crowding in the nuclear space due to the high protein expression: the reduced intermolecular distances should, thus, lead to the random proximity of components that favored the formation of PLA spots. To counteract this hypothesis, we analyzed the behavior of the 53BP1–p21 putative complex (Figure 5, Panel A). The average number of detected PLA spots grew 6 h after irradiation, whereas its value was significantly reduced at the 24-h time-point, even if the average content of the two proteins was still high, and p21, similarly to p53, presented a highly diffused localization in the nucleoplasm without evidence of local accumulation. The observed trend for the 53BP1–p21 PLA spots perfectly correlated with the DDR activity measured by γH2A.X: at 6 h, the interaction was maximum and localized at DD foci, in perfect agreement with the DNA repair activity exerted by p21 [34], whereas at 24 h, the reduced number of DDR foci led to the dramatic reduction in the number of PLA spots per cell.

The recruitment of 53BP1 to γH2A.X foci may suggest a potential involvement of p53 in the recognition and processing of DD. We, thus, performed a PLA analysis measuring p53 proximity to phosphorylated H2A.X (Figure 5 Panel A). A low, but constant, number of PLA foci were detected both 6 and 24 h after the irradiation, suggesting that the presence of a putative complex in DDR foci was possible because of a low-frequency interaction.

The resulting scenario, thus, suggested that the 53BP1–p53 interaction may be principally exerted outside the IR foci: 53BP1 was consequently requested in proximity of DSBs to support damage recognition and processing, and, maybe in a competing way, in a wide variety of nuclear loci to help p53 activity.

Focusing on this molecular interaction requires a precise spatial intracellular mapping with increased resolution.

First, to validate the results derived from the widefield image cytometry analysis, we employed the re-localization feature of our developed software. Starting from the XY-position of every cell in each image, and using the acquisition metadata associated to the same image, it is possible to re-calculate the microscope stage coordinates. We, thus, (*i*) isolated cell-subpopulations according to the p53 content and level of DD, (*ii*) created a position list, and (*iii*) re-acquired confocal stacks of the targeted cells for a 3D high-resolution view (Figure 6). The observed three-dimensional distributions confirmed that in 53BP1–p53 PLA spots, at the first time-point, localized in proximity of γH2A.X foci, but 24 h after irradiation, the 53BP1–p53 interaction was mainly localized in the nucleoplasm, with only some residual points lying close to DD spots. Comparable results were generated from the 3D analysis of the relative spatial positioning of the 53BP1–p21 PLA spots versus DDR foci (Appendix A) 6 h after the exposure to X-rays.

To better understand the obtained results, we finally focused on the quantification and localization of a putative 53BP1–p53 complex at a spatial resolution higher than the diffraction limit, and closer to the molecular scale. Single-Molecule Localization Microscopy (SMLM) allows measuring the spatial position of one or more molecular species with a precision down to 10–20 nm [19,20,21]. We applied a recently published method [35] to evaluate the colocalization of STORM data among p53 and 53BP1 (Figure 7) in the cell nucleoplasm. The calculated colocalized fractions (Figure 7, panel B–D) are perfectly compatible with the range values measured for in-vivo interactions, thus validating the presence of the putative complex.

The analysis of the p53 presence inside DD foci required a different analysis protocol, which we obtained by adapting the tools previously employed in the image cytometry pipeline (Figure 7; Panel E,F). An object-based analysis allowed segmentation of 53BP1 spots in the images generated by SMLM data, and counting of the p53 events contained therein. Comparable values for the amount of p53 were measured at both 6 and 24 h, with a slight increase in correspondence of the first time-point.

In conclusion, the measurements of the PLA assays together with the SMLM analysis confirmed a limited presence of p53 inside or in proximity of DD foci.

According to the PLA analysis, the large accumulation of 53BP1 protein in γH2A.X foci at 6 h was responsible for the formation of PLA spots in their proximity, whereas the reduced density of 53BP1 in the nucleoplasm led to a smaller number of interaction spots per cell. On the other hand, 24 h after the irradiation, the situation was completely reversed: the reduced number of 53BP1 foci and the increased diffused component caused the disappearance of the PLA spots lying close to DD spots, shifting the interaction towards the nucleoplasm space. STORM analysis allowed unmasking at a molecular scale of this concentration-dependent effect of the PLA: the correlation degree and the p53 counts were all comparable when time went on, indicating an interaction with a low, but reproducible, frequency in DD hotspots, whereas 53BP1–p53 complexes mainly localized to the sites of p53 activity with a remarkable coefficient of localization at the nanometer level.

## 3. Material and Methods

### 3.1. Cell Culture

MCF10A cells were grown in 50% Dmem High Glucose with stable L-glutamin (DMEM) (Euroclone, Milan, Italy) + 50% Ham’s F12 Medium (ThermoFisher Scientific, Waltham, MA, USA) containing 5% Horse Serum, 50 ng/mL Penicillin/Streptomycin (both from Euroclone), Cholera Toxin (Merck Life Science, Milan, Italy), 10 μg/mL Insulin (Merck Life Science, Milan, Italy), 500 ng/mL Hydrocortisone (Merck Life Science, Milan, Italy), and 20 ng/mL EGF (Pepro Tech, Cranbury, NJ, USA) at 37 °C in 5% CO_2_. Cells were grown on glass coverslips or on glass-bottom culture dishes (MatTek, Ashland, MA, USA) coated with 0.5% (wt/vol) gelatin in PBS. When the cells reached 70% confluence, they were incubated with Ethinyl-deoxyuridine (EdU) (ThermoFisher Scientific, Waltham, MA, USA), which was added to the culture media at a 10 μM final concentration, 20 min before fixation. Control cells were fixed for 10 min in 4% paraformaldehyde (wt/vol) to guarantee exponential growth, whereas treated cells were subjected to 5-Gy irradiation by an X-ray machine, and were fixed at the indicated time-points.

### 3.2. EdU Staining and Immunofluorescence of MCF10A Cells

Fixed MCF10A cells were washed and permeabilized for 10 min in a buffer containing 0.1% Triton X-100 (vol/vol) in PBS. EdU incorporation into DNA was detected using the Click-iT EdU Pacific Blue Imaging kit (ThermoFisher Scientific, Waltham, MA, USA) according to the manufacturer’s instructions. All the steps of the Click-iT reaction were performed at RT. After EdU detection, samples were incubated for 30 min in a blocking solution, 5% BSA (wt/vol) in PBS, and then for 1 h at RT with primary antibodies in blocking solution. Samples were then washed 3 times in PBS, and incubated with secondary antibodies for 45 min. The following primary and secondary antibodies were employed: Alexa488-conjugated mouse anti phosphoH2A.X (ser39) (γH2A.X) (613406, Biolegend, San Diego, CA, USA), rabbit anti-53BP1 (ab36823, Abcam, Cambridge, UK) detected by a Pacific Orange conjugated goat anti-rabbit (P31585, ThermoFisher Scientific, Waltham, MA, USA), anti-KI67 Alexa647-conjugated (558615, BD Biosciences, Franklin Lakes, NJ, USA), mouse anti-p21 (M7202, Dako, Glostrup, Denmark) detected by an anti-mouse Cy3 AffiniPure Goat Anti-Mouse IgG1 (115-165-205, Jackson-immunoresearch, West Grove, PA, USA), anti-human p53 biotin-conjugated (DO1, ab27696, Abcam, Cambridge, UK) detected by Goat anti-biotin Antibody DyLight™ 800 Conjugated (600145098, Rockland Immunochemicals, Pottstown, PA, USA). DNA was counterstained with DAPI. Samples were then mounted in Slowfade Gold Antifade Mountant (ThermoFisher Scientific, Waltham, MA, USA).

### 3.3. In-Situ Proximity Ligation Analysis (PLA)

After EdU reaction (see below for details), samples were processed for in situ PLA using the DuoLink in situ Orange detection reagent (Sigma-Aldrich, St. Louis, MO, USA) according to the manufacturer’s instructions. Then, for complete detection of the expression levels of the targeted molecules, samples were incubated with fluorochrome conjugated antibodies. The primary antibodies employed for PLA were: rabbit anti-53BP1 (ab36823, Abcam, Cambridge, UK)/mouse anti-p53 (sc-126, Santa Cruz Biotechnologies, Dallas, TX, USA), and goat anti-p53 (DO1, Santa Cruz Biotechnologies, Dallas, TX, USA)/mouse anti-pH2AX (613402, Biolegend, San Diego, CA, USA). The secondary antibodies were: Alexa488 anti-mouse and Alexa647 anti-rabbit or anti-goat conjugated antibodies (715-545-150 and 711-606-152, Jackson-immunoresearch, West Grove, PA, USA).

### 3.4. Automated Microscopy and Image Acquisition

Images were collected by a BX61 fully-motorized Olympus fluorescence microscope controlled by Scan^R software (version 2.2.09, Olympus Corporation, Tokyo, Japan) and by an inverted Nikon Eclipse Ti2 microscope (Nikon instruments, Tokyo, Japan) equipped with a LED light source (pE-4000 CoolLED, Andover, United Kingdom) and a CMOS camera (Dual ORCA Flash 4.0 Digital CMOS camera C13440, Hamamatsu, Japan) set on a 16-bit scale detection modality. NIS Elements software (version 5.30.03, Nikon instruments, Tokyo, Japan) was used to control the system for image acquisition both in the widefield and confocal mode described below. The optimal exposure time was set per each fluorescence channel by maximizing the dynamic range and avoiding saturation. Imaging was performed by an oil-immersion, 60× 1.3 NA (Olympus Corporation, Tokyo, Japan), and a 60× Plan Apo 1.4 NA objective (Nikon instruments, Tokyo, Japan). The Nikon microscope is also equipped with an A1R confocal scanhead to acquire optically-sectioned multiplane stacks. The A.M.I.CO software was employed to analyze and select events according to the quantification of a widefield acquisition. Targeted cells were then relocated by converting image coordinates into stage actual positions using the re-localization function contained in the software. Acquisitions were sequentially performed per each fluorophore with the high-speed galvanometric mirrors at a speed of 15 frames per second (2× Average) to minimize photobleaching and collection time. The slit aperture of the spectral detection system was set to maintain an optimal Signal-to-Noise ratio avoiding crosstalk among the different channels (namely 500–530 nm for the green channel, 570–600 nm for the orange one and 660–720 nm for the far red). Pinhole size has been set to 0.8 Airy Unit for every collected channel.

### 3.5. Image Analysis (A.M.I.CO Analysis Package)

The acquired images were processed by a series of newly developed computational tools based on the Automated Microscopy for Image Cytometry (A.M.I.CO) analysis package described in [22,23,24]. The software has been adapted to process the multimodal multiresolution data created in the present work, and it is freely available upon request or available on the GitHub public repository (https://github.com/MarioFaretta/AMICO (accessed on 22 July 2022)). Since it is not possible to code to all the information required (e.g., format of the position lists for the microscope, format of the acquired images, and metadata) by different microscope brands, a customization step is required to the users for adapting the code to their set-ups.

Briefly, after the correction of background and illumination inhomogeneity, cells were segmented on the DAPI signal, obtaining the measurement of shape- (area, circularity) and fluorescence- (integrated intensity per cell, mean pixel intensity per cell) related parameters for all of the acquired channels. Sub-compartments were isolated inside nuclei by segmenting the foci/spots in the H2A.X, 53BP1, and PLA images. The number of spots, size, mean, and integrated intensity were calculated for each cell. Spots can also be analyzed as independent events without considering the cell of origin.

Measurements were stored in a database in a tab-txt file format, and analyzed by a dedicated module (A.M.I.CO Plotting) employed to produce histograms and dot plots, and to perform statistical analysis with logical gating, similarly to flow cytometry. Images of the cell inside a region can also be retrieved to obtain representative visual data.

In the performed experiments, a minimum number of at least 5000 cells were analyzed.

### 3.6. Pulse and Chase EdU Assay

The pulse and chase protocol was performed as published [29], replacing BrdU with EdU to avoid the DNA denaturation step required by BrdU. Briefly, exponentially-growing cells were incubated with EdU-containing medium for 20 min. After PBS washing, cells were fed with standard cell culture medium without the synthetic nucleotide. The first time-point was fixed just after the EdU incubation. The deriving distribution of EdU versus DNA content provided the standard cell-cycle division: actively-replicating cells incorporated EdU (EdU+), whereas EdU-negative cells (EdU-) correspond to the G1 and G2 phases and were discriminated according to their DNA content (i.e., 2N and 4N, respectively) (Figure 1, Panel A, first column).

Since EdU was administered only at the time of irradiation, it provided a permanent “historical” tag during the recovery time: EdU-positive cells were irradiated during their S phase, whereas EdU-negative ones were located in the G1 and G2. In the next time-points, EdU content maintained this function (EdU+: cells irradiated during S phase; EdU-: cells irradiated during G1 and G2 phases), and it has no relation to the cell-cycle phase. The EdU-positive and -negative cells travelled across the cell-cycle, and their position can only be identified according to their DNA content, as shown by the drawn regions in Figure 1. A total of six populations were, thus, identified according to their EdU and DNA content: EdU-positive diploid (2N+), tetraploid (4N+), and intermediate DNA content (midN+) and their EdU negative counterparts (2N−, 4N−, midN−).

As a result, the cell-cycle progression can be followed for the first cell-cycle with an estimation of the duration of the different phases. The progressive disappearance of the 4N- cells provided the duration of the G2M. The appearance of events in the midN- region measured the completion of the G1 phase. The *X*-axis shift of EdU+ cells was due to the DNA synthesis (no EdU intensity changes were present, since no free EdU was present in the medium after the first time-point), and, thus, monitor the S phase progression toward G2 (4N+ cells). When EdU-positive cells underwent the first mitosis, they reappeared as the 2N+ population with EdU content reduced to one half due to the partitioning among the two daughter cells. The enrichment of events in this region, thus, monitored the start of a second cell-cycle. A complete cell-cycle was marked by DNA-EdU distribution similar to the initial time-point (0 h), as can be inferred by the 12-h time-point reported in Figure 1.

The protocol provides the advantage of monitoring the cell-cycle-specific effects of the irradiation without any chemical or physical synchronization.

### 3.7. dStorm Imaging

Single-molecule imaging (direct STORM, dSTORM) was performed with a super-resolution Nikon N-STORM microscope configured for oblique incidence excitation (N-STORM module 2, Nikon Instruments, Tokyo, Japan). dSTORM was performed in an imaging buffer that included Buffer A (10 mM Tris (pH 8.0) + 50 mM NaCl); an oxygen-scavenging system, “GLOX” (56 mg/mL Glucose Oxidase (Sigma-Aldrich, St. Louis, MO, USA); 3.4 mg/mL Catalase (Sigma-Aldrich, St. Louis, MO, USA) Stock (17 mg/mL Catalase in dH20)); and MEA (1 M Mercaptoethylamine (Sigma-Aldrich, St. Louis, MO, USA)) Stock (77 mg MEA + 1.0 mL 0.25 N HCl). MEA solution was kept at −20 °C and used within 2–3 weeks of preparation. 53BP1 was immunostained by a rabbit anti-53BP1 (ab36823, Abcam, Cambridge, UK) detected by Alexa647-conjugated anti-rabbit antibody (711-606-152, Jackson-immunoresearch, West Grove, PA, USA), whereas p53 with a human p53 mouse monoclonal antibody (DO1, Santa Cruz Biotechnologies, Dallas, TX, USA) was detected by an anti-mouse Atto 488 (62197, Sigma-Aldrich, St. Louis, MO, USA). Alexa Fluor 647 molecules were excited by using a 647-nm laser (120 mWatts nominal power), whereas Atto 488 were excited with a 488-nm laser (70 mWatts) (LU-NV laser unit, Nikon instruments, Tokyo, Japan). A 405-nm laser (20 mWatts) was used for continuous activation of dyes (both the activator and imaging lasers are continuously on). A multi-band dichroic mirror (C-NSTORM QUAD 405/488/561/647 FILTER SET; Chroma), was combined with 488-nm and 647-nm emission filters (IDEX Health & Science, Semrock Brightline^®^, West Henrietta, NY, USA). The filter-set avoided the crosstalk between channels, also blocking fluorescence due to the 405-nm activator laser. The fluorescence emission of all channels was collected through a Nikon CFI SR Apochromat TIRF 100× oil objective (1.49 NA), and was finally detected by an ORCA Flash 4.0 Digital CMOS camera C13440 (Hamamatsu, Bridgewater, NJ, USA). The number of frames and the exposure time per channel depended on the density pattern of the immunostaining and on the dye blinking-state. In each acquisition, we recorded 15,000 frames at 20 ms/frame of exposure time per channel. The focus position was maintained by monitoring the reflection of a near-infrared light from the coverslip inner surface (Nikon Perfect Focus System (PFS)). The employed fraction of the full power of both activation and excitation lasers is dependent on the blinking efficiency of the single fluorophores and bleaching rate. Single Molecule Localization fitting was performed with an Offline N-STORM Analysis module (NIS Elements software version 5.30.03, Nikon instruments, Tokyo, Japan). Drift was corrected by the autocorrelation option of the analysis software. After the localization analysis, the reconstructed STORM images are generated. In the super-resolution image reconstruction, each molecule is represented by a Gaussian spot localized by the centroid position, with the localization precision obtained from the single-molecule fit and by an amplitude value related to the number of emitted photons. Colocalization analysis was performed according to an Image Cross Correlation Spectroscopy pipeline [35]. The object-based quantification of the p53 events described in the text was performed by the A.M.I.CO software detecting 53BP1 and p53 spots inside a cell nucleus mask obtained by the widefield imaging of the two channels, recorded before starting dSTORM acquisitions, and computing the relative distance distributions thanks to the procedure coded in the software.

## 4. Conclusions

The activation of the molecular machinery devoted to the recognition and repair of DSBs, the major threats to the genome integrity, and the parallel control of the cellular physiology involve dramatic rearrangements in the chromatin structure and in the composition of molecular complexes recruited on targeted genomic loci.

The development of new promising therapies based on the employment of particle- instead of photon-radiation created the need to characterize the spatial distribution of DD induced by the deposited energy and the consequent response activated by the cells [36]. Moreover, the intricate cell-to-cell communication network established between the tumor and the surrounding host environment, and how it can be activated and modified in response to irradiation, requires maintaining a multicellular monitoring of the resulting events [37,38].

Even if the genomic revolution delivered us tools able to capture modifications of DNA up to the single nucleotide, including the mapping of DNA breaks [39,40,41,42], they completely lose the single-cell resolution, making them not optimal in approaching the heterogeneity of cells and events at the base of tumorigenesis, tumor growth, and the response to therapy. On the other hand, flow cytometry can be successfully employed to measure the number of biomolecules with single-cell resolution and optimal statistical sampling, but it does not possess the ability to enter the intracellular environment. Fluorescence microscopy can potentially give an incomparable spatial resolution useful to address the above-mentioned questions. The employment of automation to create image cytometry protocols for the collection and analysis of a large number of events can, thus, provide new tools to monitor the network of molecular interactions with a diffraction-limited resolution and more.

We presented here an example of an application of such tools applied to the study of the response to photon irradiation. We demonstrated how a large statistical sampling coupled to a high content (up to seven fluorescence parameters simultaneously) and high resolution can reveal unexpected molecular interactions to unmask new functions of the players in the DDR, i.e., 53BP1 and p53, and how they are regulated by modulating the spatiotemporal localization in the cells. Besides showing an alternative function for the 53BP1 DDR protein in helping the p53 action to regulate cell fate, the results evidenced a putative participation of the same p53 molecule in the recognition and repair of DD. A very recent paper [43] demonstrated the recruitment of p53 into γH2A.X foci for a transcription-independent function in an osteosarcoma cell model (U2OS). Our data confirm a potential direct involvement of the guardian of the genome in the DDR in a non-transformed cell model, thus including p53 in the targets worthy of a more detailed analysis for characterizing the cell response to a different type of radiation.

One of the main remaining obstacles in the diffusion of high-resolution image cytometry resides in the volume of generated data. We proposed here an approach based on the concept of a scalable resolution according to the targeted question. We provided an example of how microscope metadata can be employed to re-localize cells of interest for a second round of acquisition at the highest resolution on restricted subpopulations, e.g., to pass from a two- to a three-dimensional analysis. This way, the amount of collected data can be dramatically cut, restricting the image collection only to informative events. In the future, artificial intelligence with increasingly sophisticated machine learning algorithms can be inserted into automated microscopy frameworks to achieve an analysis-driven acquisition able to bypass today’s limitations, making image cytometry by automated fluorescence microscopy a powerful high-content high-resolution discovery tool for population studies.

## Figures and Tables

**Figure 1 ijms-23-10193-f001:**
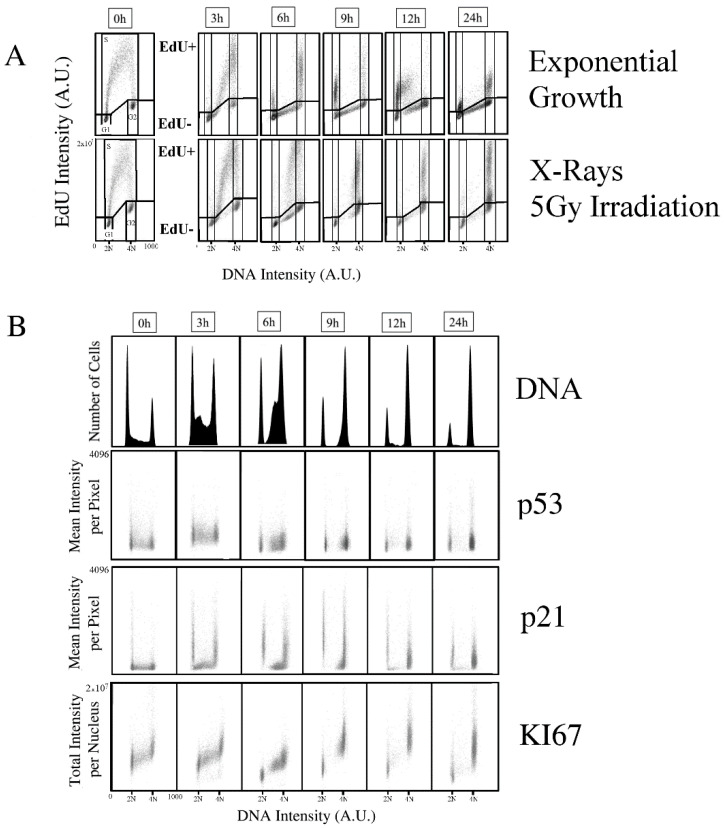
Image cytometry analysis of the cell-cycle progression after X-ray irradiation. (**A**) Pulse and chase analysis applied to control and irradiated cells after a pulse of EdU followed by washing and chasing with fresh medium. At time 0-h, only actively replicating cells incorporate (EdU+) the synthetic analogue during the 20 min of incubation. The evolution of the different populations was then followed by measurement of the DNA content (see Material and Methods). (**B**) Histograms report the DNA content distribution of the entire cell population (*n* > 5000). Dot plots report the expression profile of p53, p21, and KI67 in relation to the DNA content at the indicated time after the irradiation. Reported data refer to a representative experiment.

**Figure 2 ijms-23-10193-f002:**
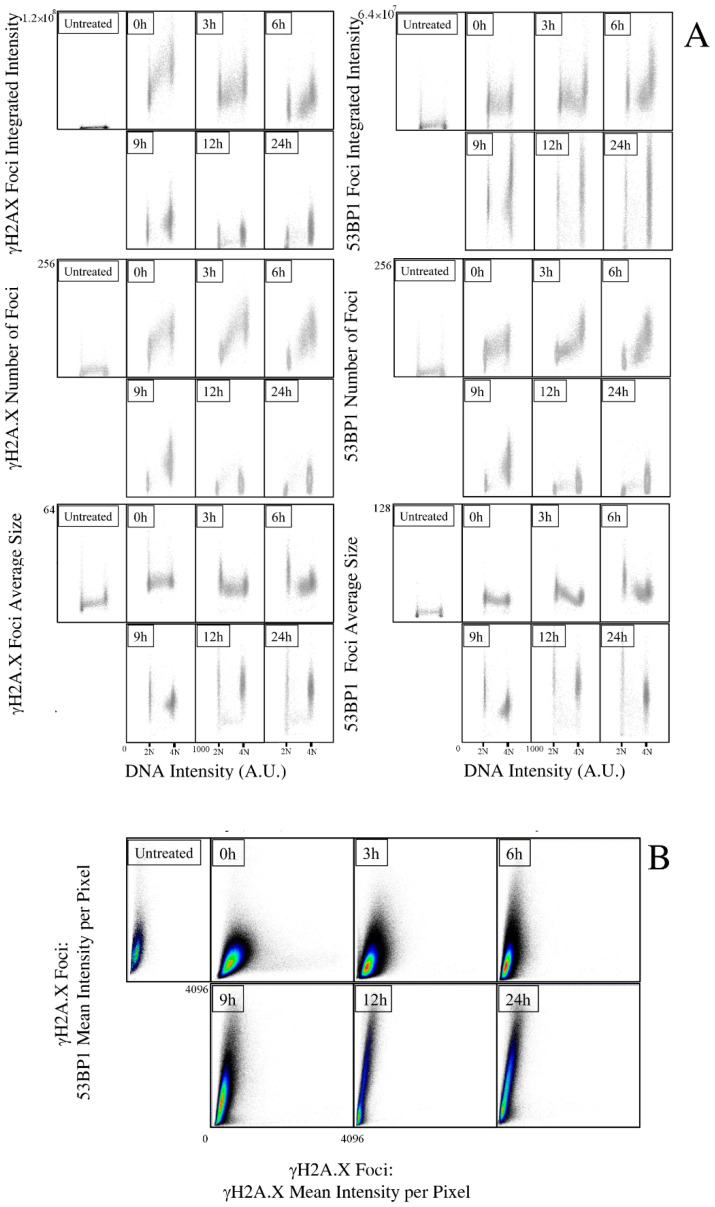
Image cytometry analysis of the IR-induced foci kinetics after X-ray irradiation. (**A**) After cell identification, γH2A.X (left) and 53BP1 (right) signals were segmented in each detected nucleus to locate and count foci, and measure their integrated intensity per nucleus (i.e., sum of the intensity of all the foci present in a nucleus) and their average size. (**B**) Statistical analysis of the foci intensity distribution. The entire population of the indicated foci (*n* > 10^5^) was analyzed as an independent entity without considering the cell of origin. The relative distribution of the two parameters reveals the kinetics of recruitment of 53BP1 protein into γH2A.X foci. Reported data refer to a representative experiment.

**Figure 3 ijms-23-10193-f003:**
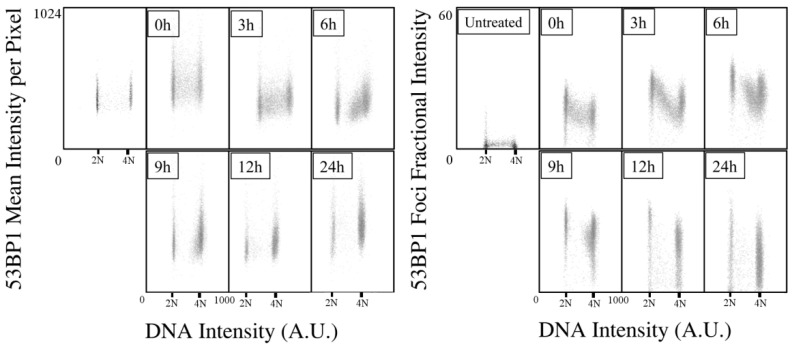
Image cytometry analysis of the IR-induced 53BP1 content and foci kinetics after X-ray irradiation. Analysis of protein content per nucleus, and of the fraction of the 53BP1 intensity localized in foci versus the total fluorescence of the nucleus in relation to the DNA content at the indicated time-points.

**Figure 4 ijms-23-10193-f004:**
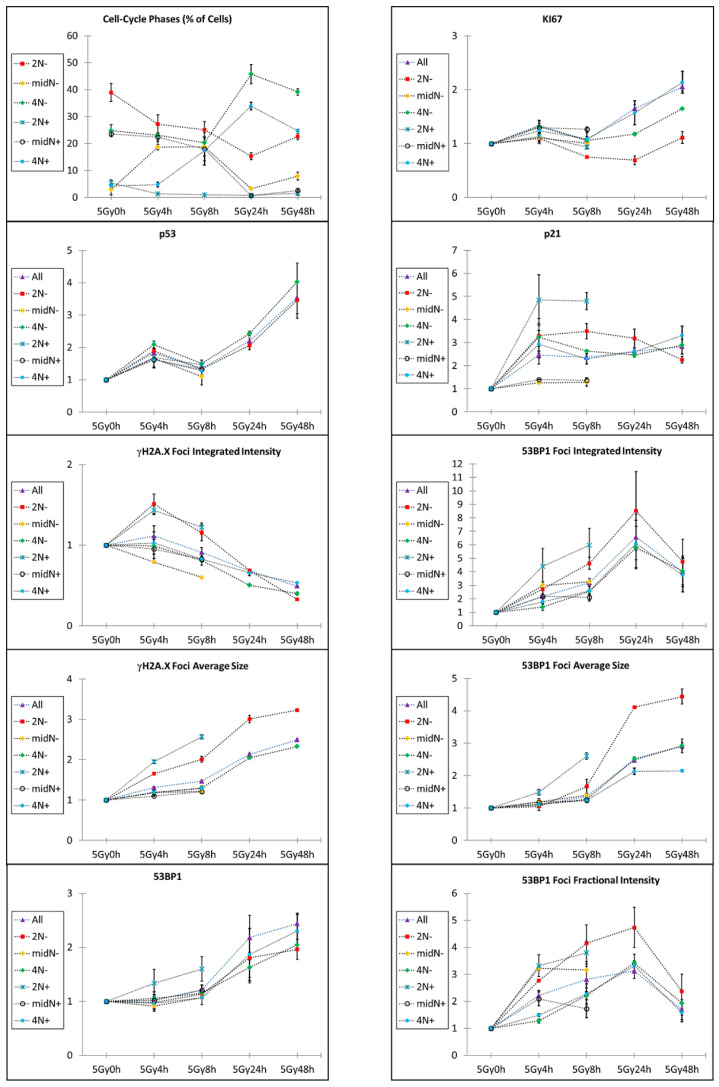
Cell-cycle analysis of the checkpoint protein content and DDR-related parameters after 5Gy irradiation. Statistical analysis of the indicated parameters calculated by targeting selected subpopulations according to their cell-cycle position at the irradiation (EdU+ and −) and at the selected time-point (DNA content: 2N, midN, 4N). The adopted regions are indicated in Figure 1. At the later time-points, some cell-cycle fractions were not indicated due to their low representativeness (number of events less than 500) caused by the cell-cycle arrest.

**Figure 5 ijms-23-10193-f005:**
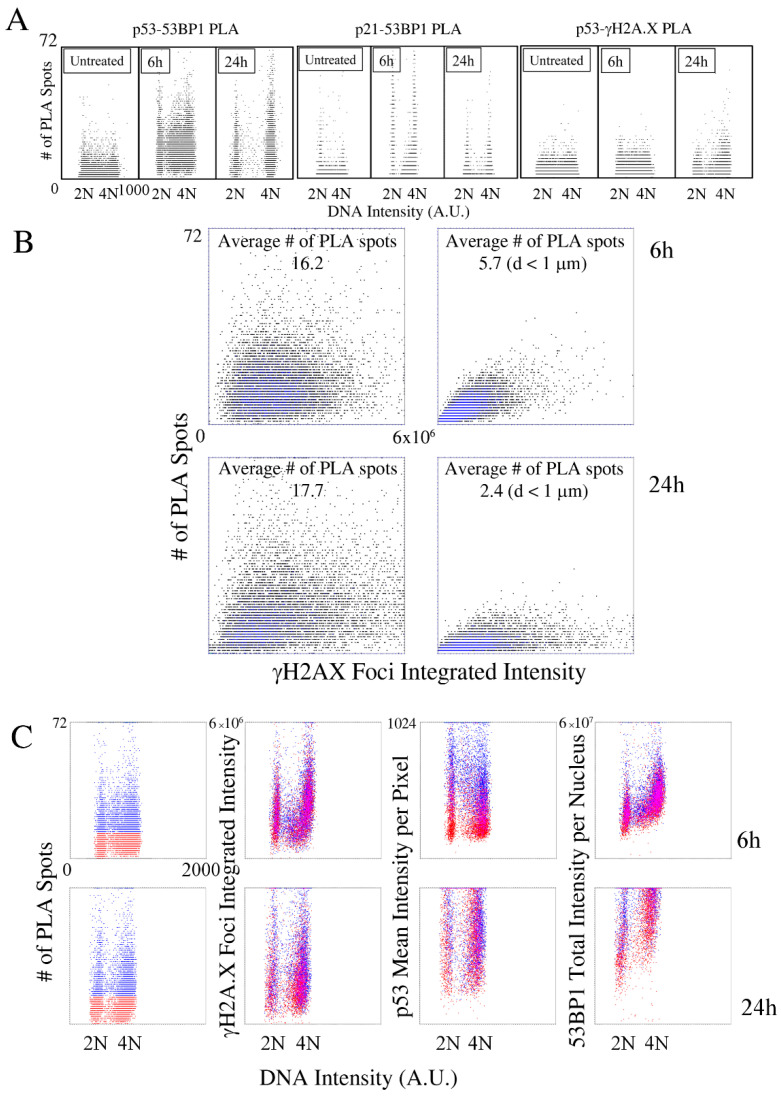
Image cytometry analysis of protein–protein interaction after X-ray irradiation detected a by Proximity Ligation Assay (PLA). (**A**) The dot-plots report the bivariate distribution of DNA content and of the number of interaction spots detected by a PLA assay between the indicated proteins. (**B**) PLA-spot population analysis (independent from the cell of origin) in relation to the position of IR foci. In the dot-plots on the right, only PLA-spots–IR-foci residing within a distance of 1 μm were considered. (**C**) Analysis of p53-, 53BP1-, and γH2A.X-related parameters according to the intensity of the 53BP1–p53 interaction. The cell population was subdivided according to the value of the median of the number of PLA spots per cell distribution.

**Figure 6 ijms-23-10193-f006:**
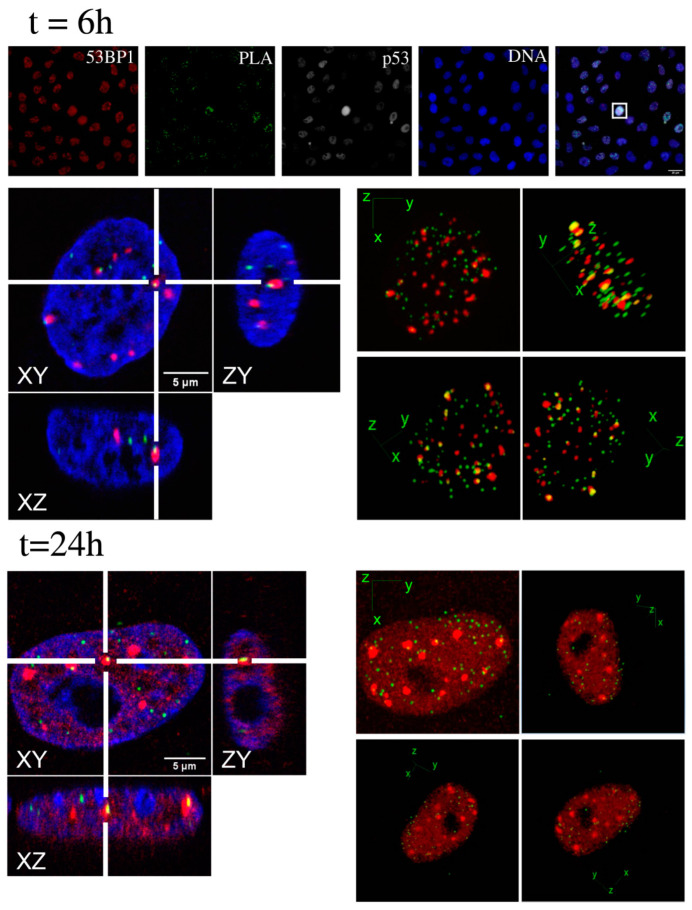
Three-dimensional high-resolution confocal microscopy of 53BP1-p53 putative complex. Samples stained for the detection of 53BP1–p53 PLA spots were analyzed according to the described image cytometry procedure to select a PLA-enriched high-p53 expression phenotype. Cells were relocated (an exemplificative cell of interest is reported (white square) in the widefield images in the upper row; scale bar, 25 microns) to perform the high-resolution 3D analysis in confocal imaging. Pictures show conventional and lateral views of a selected slice (left) and 3D maximum intensity projections from different angles (right) at the indicated time-points for a representative cell.

**Figure 7 ijms-23-10193-f007:**
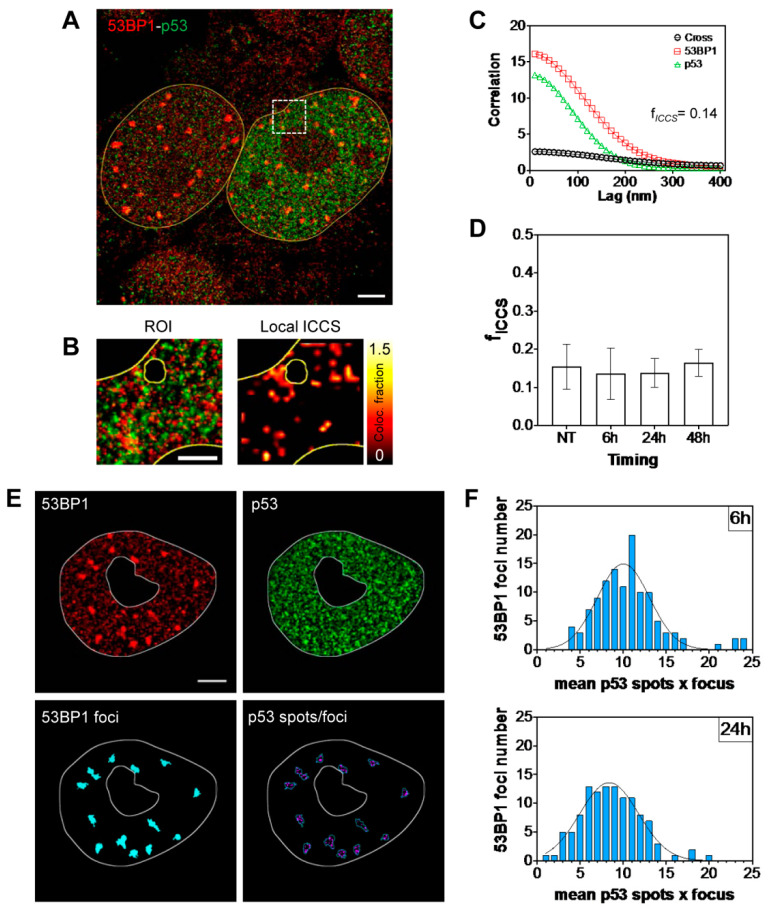
Analysis of single molecule colocalization between 53BP1 and p53. Analysis of representative dSTORM images of MCF10A nuclei acquired upon labeling of (**A**,**E**) 53BP1 (red) and p53 (green). (**B**) Shown are (from left to right) the dual-color STORM image ROI at different spatial resolutions (10 nm and 50 nm) and the map of the colocalized fraction recovered by local ICCS. (**C**) ROI spatial correlation functions recovered by ICCS. (**D**) Colocalized fraction (fICCS) extracted from ICCS analysis at different timepoints. (data are mean ± s.d. of the mean values of fICCS calculated on each cell of NT, 6 h, 24 h, and 48 h; *n* = 50). The ICCS plot shows the cross-correlation function (black squares) and the red (red circles) and green (green triangles) channel autocorrelation functions along with the corresponding fits (solid lines). (**E**) A.M.I.CO image analysis of p53 spot distribution at 53BP1 foci (cyan) on representative dual color dSTORM image of MCF10A nucleus. (**F**) Total distribution of the number of 53BP1 foci containing the number of p53 spots at 6 h and 24 h. Scale bar: 3 μm. Scale bar ROI: 1 μm.

## Data Availability

Raw data are available upon request.

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
