# Peer review of "From Double-Strand Break Recognition to Cell-Cycle Checkpoint Activation: High Content and Resolution Image Cytometry Unmasks 53BP1 Multiple Roles in DNA Damage Response and p53 Action"

_ijms, 2022, doi:10.3390/ijms231710193_

Round 1

Reviewer 1 Report

The authors are kindly requested to consider the following recommendations for the improvement of the manuscript:

-- carefully check the entire manuscript for misspelt words;

- under the section Introduction, consider highlighting the advantages of the proposed method compared to other already published methods;

- evidentiate similarly the name of the Ox axis for all figures

- In figure 4, not all figures have the Ox title

Reviewer 2 Report

In the present manuscript the authors analyze the dynamics of 53BP1 and p53, along with other DDR and cell cycle progression markers, with a high content resolution Image Cytometry methodology, unveiling new roles of 53BP1 in DDR and p53 regulation. The manuscript is interesting, the methodology applied appears very powerful and the conclusions are sound. Nevertheless, the statistic is missing in all the experiments, the data presentation needs an improvement and in some points the conclusions are not supported by the data. The authors have to address the following points to make the paper suitable for publication in MDPI.

Major points

Figure 1. The graphs are not easy to follow, probably a multicolored dot plot distinguishing EdU positive from negative would help interpretation, the author could also label control and irradiated samples to help the readers. All the presented plots should report the scale and the values on the axes. Finally, why the cell cycle profile is shown for the untreated 0h only? Can the author try to uniform all the time points?

In the figure legend the authors state that the picture report one representative experiment. How many experimental replicates have been performed? A quantitative analysis of the data must be performed based on at least three independent experiments and showing the mean and standard deviation. The same for the panel B. Here the irradiated sample only has been shown. What about the control? In the supplemental figure S1 at least Ki67 and p53 show a detectable signal in the control at each time point.

Figure 2 and 3. The authors need to show the quantitative analysis of the dot plot presented. As above, the graphs must show scale and values on the axes.

Figure 4 is very confusing. In the panel “cell cycle Phases” it is not clear why EdU administration would change so much the cell cycle distribution after irradiation. In all the other panels, I don’t understand why it is necessary to distinguish EdU positive from negative populations. Here the point should be to analyze contemporarily the accumulation trend of p21, p53 and Ki67during DDR activation and resolution. In addition, some curves are not complete, some data sets are missing. I strongly suggest simplifying the figure to make the message as clearer as possible.

Figure 5: Panel A: quantification of results is needed (histograms or box plot could be used)

Panel B: Representative images of the PLA and DDR spots localization is required

Panel C: it is hard to understand what this data means. What I understand from the picture is that the two populations (high or low p53-53BP1 complexes) are not differently distributed in the cell cycle phases or upon DDR activation. The difference between upper and lower panels remains obscure. I guess that p53 and 53BP1 abundance have an influence of PLA spots number per cells (how expected) but it is not clear how this can be evicted from the graph. The authors must make the effort to explain their data in a clearer way.

Figure 6: 53BP1 in the first panel should be gH2AX. The authors must score the number of colocalizations between PLA and gH2AX spots in the MIP to quantify the decrease of interaction between the p53-53BP1complexes and DDR sites over time, instead of simply showing a single representative cell for each condition.

Figure 7 : The authors cannot show quantification of a single representative nucleus per condition. The analysis must be extended to at least 50 nuclei per condition to be representative of the whole sample.

Supplemental figure 2: p53 label in the third panel should be p21

Raw 257-260: the author cannot conclude that “53BP1 was requested to support p53 activity with a function distinct from DDR” from the simple observation of the trends of accumulation of the proteins. This sentence must be removed

Raw 278: “While 53BP1 recruitment to IR foci provided a positive control of the PLA (data not shown)” these data must be shown. The authors need to show representative IF images of PLA in control and irradiated conditions to prove that the number of spots is coherent between the two methodologies.

Minor points

Raw 228: “…identified to identify…” please avoid repetition

The authors order displayed in the submission is different from the one in the manuscript

Reviewer 3 Report

The authors examined the p53-53BP1 interaction induced by IR with the recently developed Single-Molecular localization microscope in the manuscript. This new strategy overcomes some obstacles to generate large amount of data with high resolution. It may become a very useful tool to study protein-protein interaction and DNA damage response.

Author Response

We thank the reviewer for the positive evaluation of our work